# Universal fine grained asymptotics of free and weakly coupled Quantum Field Theory

**Weiguang Cao**[1,2]**, Tom Melia**[2]**, Sridip Pal**[∞]

[1]*Department of Physics, Graduate School of Science, The University of Tokyo, Tokyo 113-0033, Japan*

[2]*Kavli Institute for the Physics and Mathematics of the Universe (WPI), UTIAS, The University of Tokyo, Kashiwa, Chiba 277-8583, Japan*

[∞]*School of Natural Sciences, Institute for Advanced Study, Princeton, NJ 08540, U.S.A.*

*E-mail:* weiguang.cao@ipmu.jp, tom.melia@ipmu.jp, sridip@ias.edu

ABSTRACT: We give a rigorous proof that in any free quantum field theory with a finite group global symmetry G, on a compact spatial manifold, at sufficiently high energy, the density of states $\rho_\alpha(E)$ for each irreducible representation $\alpha$ of G obeys a universal formula as conjectured by Harlow and Ooguri. We further prove that this continues to hold in a weakly coupled quantum field theory, given an appropriate scaling of the coupling with temperature. This generalizes similar results that were previously obtained in $(1+1)$-D to higher spacetime dimension. We discuss the role of averaging in the density of states, and we compare and contrast with the case of continuous group G, where we prove a universal, albeit different, behavior.

## 1 Results and Summary

Black holes are absorbing objects to study. Several universal results have been obtained regarding the density of states of black holes in AdS which, via the AdS-CFT correspondence, get reflected in the high energy density of states of some dual CFT living on the boundary [1, 2]. From the perspective of the boundary CFT, specifically in two dimensions, a plethora of results are available [3–8] along with their more rigorous cousins [9–15]. Results that generalize free theory asymptotics to general spacetime dimensions with arbitrary field content and continuous symmetry groups (including projections to *e.g.* singlets) have recently been obtained using Hilbert series and Meinardus' theorem [16]. Recently, Harlow and Ooguri obtained a finer result for the density of the states transforming under some particular irreducible representation (irrep) of a finite group G, using Euclidean gravity coupled to a finite gauge field in any dimension [17], see [18] for background. Based on the generality of their argument, they conjectured

**Conjecture 1.1** (HO conjecture[17])**.** *In any quantum field theory (QFT) with a finite group global symmetry* G*, on a compact spatial manifold, at sufficiently high energy, the density of states* $\rho_\alpha(E)$ *for each irrep* $\alpha$ *of* G *obeys*

$$\rho_\alpha(E) = \frac{\dim(\alpha)^2}{|\mathrm{G}|}\rho(E)\,, \tag{1.1}$$

*where* $\rho(E)$ *is the asymptotic density of states irrespective which irrep the states fall into,* $|\mathrm{G}|$ *is the order of the group* G *and* $\dim(\alpha)$ *is the dimension of the irrep* $\alpha$.

Since the density of states is a distribution, a more precise formulation of the conjecture requires us to formulate conjecture 1.1 in a way where the density of states is integrated against a suitable function, for example the Boltzman weight, leading to a statement in a canonical ensemble, or some compactly supported function, leading to a statement in a microcanonical ensemble.

To this end, let us define the partition functions

$$Z_\alpha(\beta) = \int d\Delta \, \rho_\alpha(\Delta) e^{-\beta\Delta}, \quad Z(\beta) = \int d\Delta \, \rho(\Delta) e^{-\beta\Delta}. \tag{1.2}$$

Eq. (1.1) implies that for a finite group in the $\beta \to 0$ limit,

$$Z_\alpha(\beta) = \frac{\dim(\alpha)^2}{|G|} Z(\beta). \tag{1.3}$$

The rationale behind treating the microcanonical and canonical version separately is that, beyond the free theory, it is often technically hard to prove the equivalence between the microcanonical and the canonical ensemble beyond the leading order behavior of thermodynamically relevant quantities like energy or entropy. On the other hand, we note that the HO conjecture 1.1, when stated as a statement about entropy (proportional to log of density of states) precisely probes an order one number in the difference between $\log \rho_\alpha$ and $\log \rho$, where both quantities are individually expected to behave extensively. This necessitates stating the theorems in both a canonical and a microcanonical version; the level of rigor and the strength of the assumptions vary when we analyse the respective versions for a weakly interacting theory below.

The precise version of the conjecture 1.1 can be stated in one of the following ways.

**Conjecture 1.2.** *Consider a QFT with a finite group global symmetry* G *with order* $|G|$, *on a compact spatial manifold. The irreps of* G *are labelled by* $\alpha$ *and* $\dim(\alpha)$ *denotes the dimension of irrep* $\alpha$.

*(i) At sufficiently small* $\beta$, *the reduced partiton function* $Z_\alpha(\beta)$ *of such a theory, constructed out of the irrep* $\alpha$ *of* G *obeys*

$$Z_\alpha(\beta) = \frac{\dim(\alpha)^2}{|G|} Z(\beta), \tag{1.4}$$

*where* $Z(\beta)$ *is the full partition function, taking into account all the states.*

*(ii) At sufficiently high energy, the density of states* $\rho_\alpha(E)$ *for each irrep* $\alpha$ *of* G *obeys*

$$\int_{E-\delta}^{E+\delta} dE' \, \rho_\alpha(E') = \frac{\dim(\alpha)^2}{|G|} \int_{E-\delta}^{E+\delta} dE' \, \rho(E'), \tag{1.5}$$

*where* $\delta$ *is a suitable order one number, and* $\rho(E)$ *is the asymptotic density of states irrespective which irrep the states fall into.*

For $(1+1)$-D both versions have been proven in [14]. In this short note, we show that

**Theorem 1.3** (Canonical Version, Free). *Consider a free QFT on a compact spatial manifold $\mathcal{M}$, with a finite group global symmetry* G *such that the action of the group* G *is faithful in the sense that there is a notion of a fundamental field that transforms in a faithful irrep of the group* G.

(i) *If $\mathcal{M} = S^{d-1}$, for a sufficiently small $\beta$, we have*

$$Z_\alpha(\beta) = \frac{\dim(\alpha)^2}{|\mathrm{G}|} Z(\beta) \,. \tag{1.6}$$

(ii) *The high temperature relation given by eq. (1.6) is true even if we replace the spatial manifold $S^{d-1}$ with any arbitrary compact spatial manifold $\mathcal{M}$.*

The above theorem is the main result in this paper. We remark that for a free theory, the spectrum is regularly spaced and completely known. Thus one can go trivially from a canonical ensemble to a microcanonical one. Indeed, for free theory one can sensibly talk about the HO conjecture 1.1 with the density of states replaced by the actual degeneracy of states at some sufficiently high energy.

Now we turn our attention to the weakly coupled QFTs. If we perturb a CFT by a relevant deformation with energy scale $\mu$, the theory flows away from the fixed point. However at high temperature i.e $T \gg \mu$ , the physics is controlled by the UV fixed point. Therefore, in the weak coupling limit, we expect the result relevant for the UV fixed point to continue to be true. In the present context, this implies that we can leverage the result that we proved for free CFT to deduce the same for weakly coupled QFT (see [19] as well).

Naively one would expect that

$$Z(\beta) = Z_{\text{Free}}(\beta) \left[1 + O(\lambda)\right] \,, \tag{1.7}$$

where $\lambda$ is the coupling parameter (appropriately made dimensionless by a relevant energy scale, for example $\mu$). While the above equation is true for finite $\beta$, the subtlety lies in the fact that the order of $\lambda$ estimate is not necessarily uniform in $\beta$. In other words, the correction term $O(\lambda)$ can potentially depend on $\beta$ and might get enhanced as we take the $\beta \to 0$ limit. This necessitates that we take a simultaneous limit $\beta \to 0$ and $\lambda \to 0$ i.e we are scanning over a sequence of weakly coupled theories such that they become arbitrarily weak as $\beta \to 0$.

To this end, we consider a sequence of such theories labeled by $Q_\lambda$ such that $\lambda \to 0$. In this notation $Q_0$ is the free theory. The energy spectrum of the theory $Q_\lambda$ is given by $E(\lambda)$ with density of states $\rho(\lambda, E)$. We further require that at a given energy $E$, we can make $\lambda$ sufficiently small such that $\rho(\lambda, E')$ has a perturbative description in $\lambda$ for $E' < E$ and this continues to hold as we let $E \to \infty$ with possibly tuning $\lambda \to 0$. We will call such a sequence **T**. The sequence **T** can as well be a sequence of free theories. From now on by weakly coupled theory, we will mean the elements of the sequence **T**.

**Theorem 1.4** (Microcanonical, Weakly coupled)**.** *Consider the sequence* **T** *of weakly coupled QFTs with a finite group global symmetry* G*, on a compact spatial manifold* $\mathcal{M}$*. We assume that the action of the group* G *is faithful in the following sense: in a weakly coupled description there is a notion of a fundamental field that transforms in a faithful irrep of the group* G*.*

*If* $\mathcal{M} = S^{d-1}$*, for a given high energy* $E$*, for sufficiently small* $\lambda$*, the density of states* $\rho_\alpha(\lambda, E)$ *for each irrep* $\alpha$ *of* G *obeys*

$$\int_{E-\delta}^{E+\delta} \mathrm{d}E' \, \rho_\alpha(\lambda, E') = \frac{\dim(\alpha)^2}{|\mathrm{G}|} \int_{E-\delta}^{E+\delta} \mathrm{d}E' \, \rho(\lambda, E') \,, \tag{1.8}$$

*where* $\delta$ *is a suitable order one number,* $\rho(\lambda, E)$ *is the density of states irrespective which irrep the states fall into,* $\rho_\alpha(\lambda, E)$ *is the density of states restricted to irrep* $\alpha$ *of the group* G*.* $|\mathrm{G}|$ *is the order of the group* G *and* $\dim(\alpha)$ *is the dimension of the irrep* $\alpha$*.*

Under the assumption that the high temperature asymptotics of partition function of weakly coupled free theory for sufficiently small coupling can be captured by the Plethystic exponential, one can arrive at the canonical version of the above theorem. We state this in appendix. A. We further detail the argument and assumptions under which the leading behaviour of the canonical partition function is unchanged in passing to a weakly coupled interacting QFT from a free QFT.

The organization of the paper is as follows. In Section 2, we show how theorems 1.3.(i) and 1.3.(ii) follow from the Plethystic form of the partition function of the free theory. In Section 3 we expound upon weakly coupled theory and prove the microcanonical theorem 1.4. In Section 4 we conclude with a discussion that includes the relevance of smearing in stating formulas for the asymptotic density of states, and comparisons of the results here to the case of continuous symmetries. Symmetry resolution of weakly coupled free field theory in the canonical ensemble is discussed in appendix A.

**Note added:** After this work had been completed and was in preparation, Ref. [20] was posted as a preprint, which takes a complementary, quantum information based approach to proving conjecture 1.1.

## 2 Free CFT

In this section we prove the theorem 1.3.(i) and then extend it to prove theorem 1.3.(ii). The microcanonical version as given by theorem 1.4 restricted to the free theory follows from performing a Fourier transformation, in the $\tau = i\beta$ variable, of the high temperature partition function. Here we can make $\delta = 0$ i.e we don't need to smear the density of states, rather we can sensibly talk about degeneracy of states at physical spectrum since the spectrum of a free theory is regularly spaced and completely known. Key to our proof is the theorem 6.2 in [21] and its suitable variations, as explored in [16].

To prove theorem 1.3.(i), we begin by working with the case of a free scalar field $\phi$ transforming in the $\gamma$ irrep of the group G. If $\gamma$ is a complex irrep, we should also include the $\phi^\dagger$, transforming in the conjugate irrep $\bar{\gamma}$. The partition function of such a theory on $S^{d-1} \times S_\beta$ with a $g$ insertion is given by the exponentiation of single particle partition function, see *e.g.* [22]

$$Z(\beta, g) = \exp\left[\sum_{n=1}^{\infty} \frac{e^{-n\beta\Delta_\phi}}{n} \left(\chi_\gamma(g^n) + \chi_\gamma^*(g^n)\right) \frac{(1 - e^{-2n\beta})}{(1 - e^{-n\beta})^d}\right]. \tag{2.1}$$

Here $\Delta_\phi = (d-2)/2$. We note that in the $\beta \to 0$ limit, the value of $\Delta_\phi$ does not play any role. Another point worth emphasizing is that $Z(\beta, g)$ is manifestly real. We want to establish that up to exponentially suppressed corrections, we have

$$Z(\beta, g) \underset{\beta \to 0}{\simeq} Z(\beta, e)\delta_{g,e}. \tag{2.2}$$

Let us recap some basic group/character theory facts. If $\gamma$ is a nontrivial faithful irrep of the group G, for $g \in$ G, $\chi_\gamma(g) = \chi_\gamma(e)$ implies $g = e$ (where $e$ is the identity element). Thus we have for all $n \in \mathbb{N}$,

$$\left(\frac{\chi_\gamma(g^n) + \chi_\gamma^*(g^n)}{2}\right) = \mathrm{Re}\chi_\gamma(g^n) \leq |\chi_\gamma(g^n)| \leq \chi_\gamma(e), \tag{2.3}$$

where the equality is only achieved if $g^n = e$ (this is obtained by using the above mentioned fact about faithful irrep, applied to the group element $g^n$). Note that $\chi_\gamma(e) = \dim(\gamma)$. Now, starting from eq. (2.1), in the $\beta \to 0$ limit, we have

$$\log Z(\beta, g) \underset{\beta \to 0}{\simeq} 2\beta^{1-d} \sum_{n=1}^{\infty} \frac{1}{n^d} \left(\chi_\gamma(g^n) + \chi_\gamma^*(g^n)\right) \leq 4\chi_\gamma(e)\beta^{1-d}\zeta(d), \tag{2.4}$$

leading to

$$\log Z(\beta, g) \underset{\beta \to 0}{\simeq} \begin{cases} 4\beta^{1-d}\zeta(d)\dim(\gamma), & \text{If } g = e \\ 4\beta^{1-d}\zeta(d)\dim(\gamma)\left[1 - \epsilon\right], & \text{If } g \neq e, \epsilon > 0. \end{cases} \tag{2.5}$$

We see that for $g \neq e$, the partition function $Z(\beta, g)$ is exponentially suppressed by a factor of $e^{-4\beta^{1-d}\zeta(d)\dim(\gamma)\epsilon}$ compared to $Z(\beta, e)$. One might worry that $\epsilon$ can be made arbitrarily small by varying the element $g$, but this is not the case since |G| is finite; thus $Z(\beta, g \neq e)$ is indeed suppressed. We will further show that $\epsilon$ is bounded from below uniformly by a positive constant which depends on |G|, but does not depend on the specific group element $g$.

• *Bound on $\epsilon$*: For concreteness, say $\gamma$ is a non-trivial irrep and $g \neq e$ and $g$ has order $k \in \mathbb{N}$. Of course $|\mathrm{G}| \geq k \geq 2$ since $g \neq e$. Since $\gamma$ is a faithful irrep, we have

$$\left(\frac{\chi_\gamma(g) + \chi_\gamma^*(g)}{2}\right) = \sum_{i=1}^{\dim(\gamma)} \cos\left(\frac{2\pi m_i}{k}\right), \tag{2.6}$$

where the $m_i \in \mathbb{Z} \cap [0, k)$ and at least one of the $m_i \geq 1$. Let us estimate the difference between $g \neq e$ and $g = e$ coming from this term:

$$\left[2\dim(\gamma) - \left(\chi_\gamma(g) + \chi_\gamma^*(g)\right)\right] = 2 \sum_{i=1}^{\dim(\gamma)} \left[1 - \cos\left(\frac{2\pi m_i}{k}\right)\right] \geq 4\sin^2\left(\frac{\pi}{|G|}\right), \tag{2.7}$$

where we have used the fact that not all $m_i$ can be 0 *i.e.* at least one of the $m_i$ is greater than or equal to 1 and $k \leq |G|$. Thus we deduce that

$$\epsilon \geq \frac{4\sin^2\left(\frac{\pi}{|G|}\right)}{\dim(\gamma)} > 0. \tag{2.8}$$

This shows that up to an exponentially suppressed correction, we indeed have

$$Z(\beta, g) \underset{\beta \to 0}{\simeq} Z(\beta, e)\delta_{g,e}. \tag{2.9}$$

The final step uses the character orthogonality

$$Z_\alpha(\beta) = \frac{\dim(\alpha)}{|G|} \sum_{g \in G} \chi_\alpha^*(g) Z(\beta, g), \tag{2.10}$$

in the same way as used in [14, 17] and we arrive at Theorem 1.3.(i). We make few important remarks, one of which is aimed towards proving theorem 1.3.(ii) for the free theory:

1. The above argument can easily be generalized to any free field theory set up including fermionic degrees of freedom.

2. This works whenever there is a single particle or single letter partition function $I(\beta)$ and we get the full partition function (or index calculation) by exponentiating that i.e

$$Z(\beta, g) = \exp\left[\sum_{n=1}^{\infty} \frac{1}{n} \left(\chi_\gamma(g^n) + \chi_\gamma^*(g^n)\right) I(n\beta)\right]. \tag{2.11}$$

• **Proving theorem 1.3.(ii):** If we put a free theory on a compact spatial manifold, the above form remains true. In fact, one can show that [23] (see section $II$ of this paper) in the $\beta \to 0$ limit, the single letter partition function is controlled by Weyl asymptotics of growth of number of eigenvalues of Dirichlet Laplacian on that manifold. This immediately implies that $I(n\beta)$ goes like $(n\beta)^{-d+1}$. Thus we arrive at theorem 1.3.(ii).

• **Proving theorem 1.4 for free theory on compact spatial manifold:** This follows from theorem 1.3.(ii) by doing inverse Laplace transformation.

3. The argument is somewhat conceptually similar to the heuristic argument presented in [17] at the very end of the paper.

4. The result holds for weakly coupled QFT where there exists an approximate sense of exponentiation of the single particle partition function. We will make this more precise in the next section.

# 3 Weak coupling regime

In this section we prove that the smeared version of the conjecture 1.1 holds under a suitable weak coupling assumption and thereby we prove theorem 1.4.

## 3.1 Anomalous dimension and Microcanonical version

In a free theory the scaling dimension of the operators are regularly spaced. For example, in free bosonic scalar field theory in four dimensions, the gap between consecutive operator dimension is one.[1] Before we turn on interactions, there are no operators at some non-integer scaling (mass) dimension. When we turn them on, the operators obtain anomalous scaling (mass) dimensions and spread out away from the integer points.

Let $d_k$ denote the number of operators with dimension $k \in \mathbb{Z}$ for the free theory, and $d_{k,\alpha}$ the number of operators with dimension $k \in \mathbb{Z}$ and transforming in the irrep $\alpha$. As explained in the introduction, we consider the sequence $\mathbf{T}$ of weakly coupled theories. Specifically, by weak coupling we mean that the density of states $\rho(\lambda, E')$ is perturbative in $\lambda$ for $E' < E$ where $E$ is the energy of a given high energy state. In this regime, it makes sense to talk about the shift in the energy due to the coupling $\lambda$. If we insist on using the state-operator correspondence, one can phrase the shift as the anomalous dimension of the operator. In this section, while we will use scaling dimension $\Delta$ and energy $E$ interchangeably, it is not necessary to phrase everything in the language of (anomalous) scaling dimension.

If the shift is small enough, which can be arranged by making $\lambda$ sufficiently small, we must have

$$\int_0^\infty d\Delta' \, \Theta \left[ \Delta' \in (k - 1/2, k + 1/2) \right] \, \rho(\lambda, \Delta') = d_k \,, \quad k \leqslant N \tag{3.1}$$

where $\Theta \left[ \Delta' \in (k - 1/2, k + 1/2) \right]$ is the characteristic function for the open interval $(k - 1/2, k + 1/2)$. As $N \to \infty$ if needed, we might have to take the coupling $\lambda \to 0$ limit to make sure that eq. (3.1) holds true.

Eq. (3.1) further implies that

$$\int_0^\infty d\Delta' \, \Theta \left[ \Delta' \in (k - 1/2, k + 1/2) \right] \, \rho_\alpha(\lambda, \Delta') = d_{k,\alpha} \,, \tag{3.2}$$

Here $\lambda$ is the same as that appearing in eq. (3.1).

The above trivially leads to as $k \to \infty$

$$\lim_{k \to \infty} \frac{\left( \int_{k-1/2}^{k+1/2} d\Delta' \, \rho_\alpha(\lambda, \Delta') \right)}{\left( \int_{k-1/2}^{k+1/2} d\Delta' \, \rho(\lambda, \Delta') \right)} = \lim_{k \to \infty} \frac{d_{k,\alpha}}{d_k} = \frac{\dim(\alpha)^2}{|\mathrm{G}|} \,, \tag{3.3}$$

where we have used the fact that in the free theory, theorem 1.4 holds true for $\delta = 0$ with $E = n \to \infty$. This concludes the proof of theorem 1.4 for weakly coupled QFT.

---

[1] For fermionic (bosonic) theories in even (odd) space-time dimensions, the gap is half-integer. Nonetheless, the crux of the argument we present relies only on having regular spectra, and whether the gap is 1 or 1/2 is not important. Thus WLOG, we will assume the gap is 1 in free theory.

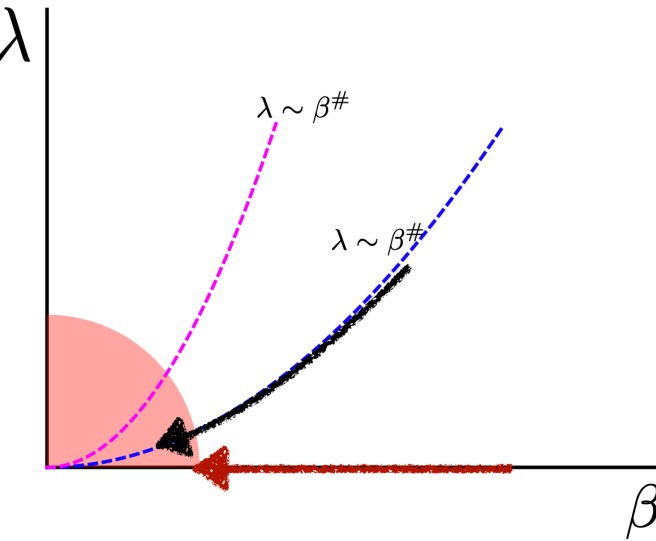

**Figure 1**. Different approaches to high temperature in the $\lambda - \beta$ plane.

Let us understand the consequences of the above analysis in more detail. Suppose at a UV fixed point we have one relevant coupling $\lambda$, and consider the $\beta \to 0$ limit in $(\beta, \lambda)$ plane, see Fig. 1. When $\lambda = 0$, we are at the UV free fixed point, and the limit is approached along the $\beta$ axis in the figure, giving the free theory high-temperature behavior. If instead $\lambda$ is fixed and again we approach $\beta \to 0$, then the high temperature behaviour changes. The above theorem implies that the anomalous dimension can not be bounded along this line. To achieve a bounded anomalous dimension, we need to consider a simultaneous limit where both $\lambda \to 0$ and $\beta \to 0$. Note this is not the same as $\lambda = 0$ and $\beta \to 0$. These are the curved flows depicted in Fig. 1.

## 3.2 Examples: Wilson-Fisher fixed point

The purpose of this subsection is to provide examples which exhibit that the anomalous dimension of heavy operators can be bounded by an order one number provided we make the coupling sufficiently small.

Consider the massless $\phi^4$ theory in $d = 4 - \epsilon$ dimension:

$$S = \int d^{4-\epsilon}x \, \left[\frac{1}{2}(\partial\phi)^2 + \frac{1}{4!}\lambda\mu^\epsilon\phi^4\right] . \tag{3.4}$$

For $\epsilon > 0$ , $\phi^4$ is a relevant coupling and the theory flows from the Gaussian free fixed point to the interacting Wilson Fisher (WF) fixed point, where coupling takes the following value

$$\lambda = \lambda_* = \frac{16\pi^2}{3}\epsilon + O(\epsilon^2) . \tag{3.5}$$

The leading order anomalous dimensions generically have the following form *e.g.* [24–26],

$$\gamma_\mathcal{O} \simeq \lambda \left(\Delta_\mathcal{O}\right)^\alpha . \tag{3.6}$$

If we want them to be bounded by an order one number, then we are required to take the coupling to be small such that $(\Delta_{\mathcal{O}})^\alpha \lambda \sim O(1)$. The theorems considered in previous subsection imply that so long as this holds true, the partition function in the high temperature limit should behave like that of the free theory. Note that operators with large scaling dimension can acquire large anomalous scaling dimension unless we scale $\lambda$ and $\beta$ and $\Delta$ appropriately. For example, we can consider the operator $\phi^n$ which has scaling dimension $\Delta = n$ at free fixed point. The anomalous dimension $\gamma_{\phi^n}$ of such an operator is given by

$$\gamma_{\phi^n} = \frac{1}{6}n(n-1)\epsilon + O(\epsilon^2), \quad n > 1. \tag{3.7}$$

Clearly if we want to bound the anomalous dimension by an order one number we would require $\epsilon \sim \Delta^{-2}$ (where at the WF fixed point, we have $\lambda = \lambda_* = O(\epsilon)$). We note that by reversing the same logic, if we knew that there *are* corrections to the free energy in the high temperature limit at fixed coupling, we would immediately conclude that for fixed finite coupling the anomalous dimensions can not be uniformly bounded *i.e.* $\alpha > 0$ given eq. (3.6). The breakdown of perturbation theory for small but fixed finite $\epsilon$ has been pointed out in [27, 28].

We can consider another example: the $O(N)$ model in $d = 3 - \epsilon$ dimensions

$$S = \int \mathrm{d}^{3-\epsilon}x \, \left( \frac{1}{2}\partial_\mu\phi_a\partial^\mu\phi_a + \frac{\lambda}{6!}\mu^{2\epsilon}\left(\phi_a\phi_a\right)^3 \right) , \quad a = 1, 2, \cdots N . \tag{3.8}$$

We focus on two kind of operators $\Phi_{2p} = (\phi_a\phi_a)^p$ and $\Phi_{2p+1} = \phi_a \left(\phi_a\phi_a\right)^p$, whose anomalous dimension are [29]

$$
\begin{aligned}
\gamma_{\Phi_{2p}} &= \frac{p(2p-2)(10p+3N-8)}{3(22+3N)}\epsilon + O(\epsilon^2) , \\
\gamma_{\Phi_{2p+1}^a} &= \frac{p(2p-1)(10p+3N+2)}{3(22+3N)}\epsilon + O(\epsilon^2) .
\end{aligned}
\tag{3.9}
$$

Here, a bounded anomalous dimension requires $\epsilon \sim \Delta^{-3}$. Again, if we knew that there are corrections to the free energy in the $\beta \to 0$ limit, we would immediately conclude that for fixed finite coupling the anomalous dimensions can not be uniformly bounded.

## 4 Discussion

**Smearing: a cautionary tale**

Strictly speaking, on a compact space, the spectrum of the theory is discrete, and the density of states makes sense only after one smears (integrates) it over a small energy window centered at some high energy. Hence, one should be careful when interpreting the $\dim(\alpha)^2/|\mathrm{G}|$ factor, especially for a weakly coupled (but not quite free) theory, where the averaged density can have some order one multiplicative correction.

The following is an illuminating interacting example from (1+1)-D CFT, which shows the importance of smearing. If a symmetry group is G, then one might naively expect a degeneracy of $\dim(\alpha)$ in the density of states for a given irrep $\alpha$, rather than something proportional to $\dim(\alpha)^2$. To be precise, the exact density of states is

$$\rho_\alpha(\Delta') = \sum_\Delta \text{degeneracy}(\Delta)\delta(\Delta' - \Delta)\,, \tag{4.1}$$

where the sum runs over the physical spectrum $\Delta$, restricted to the irrep $\alpha$. One would generically find that the degeneracy($\Delta$) is proportional to $\dim(\alpha)$. The resolution comes from the fact that the irrep $\alpha$ appears $\dim(\alpha)$ times *on average* and hence only after averaging over a sufficiently large window, one can see the $\dim(\alpha)^2$ factor. This has already been emphasized in [14] in context of $1 + 1$ D CFT, and we repeat the example given there in the below.

Consider a 3-state Potts model with central charge $c = 4/5$. This theory has a global symmetry $S_3$. There are two doublets of the primaries in the nontrivial two dimensional irrep of $S_3$. One doublet has scaling dimension $4/3$ while the other one has $2/15$. (See for example [30], Eq. 5.42, where they are denoted as $(\sigma, \sigma^*)$ and $(Z, Z^*)$). All the descendants of these primaries fall into the doublet irrep. Now if we consider an energy window of width $2\delta \gtrsim 1$, both the irreps appear and we have contribution proportional to $2 \times 2 = 4$, this is the $\dim(\alpha)^2$ factor. Now one could have chosen a very tiny window, for example a window centered at some $\Delta$ with width $1/6$ (the number is designed in a way so that $\{4/3\} - \{2/15\} > 1/6$) such that the window can not contain both the doublets, and as such the density of states is proportional to $2 = \dim(\alpha)$, rather than $\dim(\alpha)^2$.

It would be nice to understand how these subtleties appear in the recent proof [20] of the conjecture 1.1, inspired by quantum information techniques [31]. On a similar note, it is intriguing to investigate how one might see the role of averaging/smearing in the context of equipartition of entropy [32, 33] in (1 + 1)-D CFT. A plausible option is that these proofs are more geared towards the canonical formulation of the conjecture and directly applicable at the infinite temperature limit of the QFT. It would be interesting to elucidate this further.

**Generalization to continuous symmetry**

It is worth highlighting a difference in behaviour between QFTs with finite group symmetries, for which the above theorems are concerned, and QFTs with continuous symmetries. In particular, for projection to the singlet sector, we have suppression by the order of the group G for a finite group. This is in contrast to the case of a continuous Lie group as global symmetry with Lie agebra $\mathfrak{g}$. It has been shown in [16] that for free theories on spatial manifold $S^{d-1}$, in the $\beta \to 0$ limit, the reduced partition function for the singlet sector behaves

$$Z_{\text{Singlet}}(\beta) = O(1) \times \beta^{\frac{(d-1)\dim\,\mathfrak{g}}{2}} Z(\beta)\,, \tag{4.2}$$

that is, an extra suppression by a positive power of $\beta$ occurs. We refer the reader to section 5.2 of [16] for further details; an analysis in $1 + 1$-D also appears in Ref. [14], which also includes

interacting CFTs. A straightforward extension of these methods implies a suppression with the same $O(1)$ number as above

$$Z_\alpha(\beta) = O(1) \times \dim(\alpha)^2 \beta^{\frac{(d-1)\dim \mathfrak{g}}{2}} Z(\beta) \,. \tag{4.3}$$

The origin of the extra suppression factor can be understood by noting that the character orthogonality theorem involves an integral of the form:

$$Z_\alpha(\beta) = \dim(\alpha) \int d\mu_G \; \chi_\alpha^*(g) Z(\beta, g) \,, \tag{4.4}$$

The behaviour of the full partition function as the $\beta \to 0$, shown in [16] (see Section 5), is

$$Z(\beta, g) \underset{\beta \to 0}{\simeq} Z(\beta, e) e^{-\frac{f(\vec{\omega})}{\beta^{d-1}}} \,. \tag{4.5}$$

Here $\vec{\omega}$ is a dim $\mathfrak{g}$ dimensional vector, characterizing the deviation of the group element from the identity and $f(\vec{\omega})$ is a quadratic polynomial of components of $\vec{\omega}$. This factor captures the Gaussian fluctuation around the saddle $g = e$ of the integral eq. (4.4) and integrates to give the aforementioned suppression. For a finite group, only $g = e$ contributes and there is no notion of fluctuation, hence no such extra suppression occurs. In light of the above discussion, we formulate a unified version of the conjecture, which applies to both the finite and continuous group,

**Conjecture 4.1.** *In any QFT with a global symmetry* G, *on a compact spatial manifold, at sufficiently small* $\beta$, *the reduced partition function* $Z_\alpha(\beta)$, *constructed out of the irrep* $\alpha$ *of* G *obeys*

$$\frac{Z_\alpha(\beta)}{Z_{Singlet}(\beta)} = \dim(\alpha)^2 \,. \tag{4.6}$$

One can also state a microcanonical version, but we do not repeat this here.

**Three further remarks**

First, we expect the statement in the theorems 1.4 and A.3 to be true even if we replace the spatial manifold $S^{d-1}$ with an arbitrary compact spatial manifold $\mathcal{M}$; we leave a detailed analysis of this to be done elsewhere.

Second, the results of this paper could have relevance to the calculation of SUSY indices. Here, one generally use supersymmetry to argue that the index is invariant over the moduli. Then one may simply work in the weak coupling limit to obtain the index via a free theory computation. Such twisted indices have been looked at in the literature in various particular examples [34–40]. We highlight that a number of developments related to the application of Hilbert series and the Plethystic exponential in QFT appear within the context of the operator counting program in effective field theory, see in particular the technical and conceptual ideas appearing in the works [22, 41, 42], as well as several others [43–49].

Third, we can make a general remark about the scaling behaviour of partition functions with respect to $\beta$ in a generic interacting CFT/QFT, and make the connection to an intuitive but non-rigorous argument found in [50]. Consider an interacting CFT with a partition function on $S^{d-1} \times S_\beta$ written in the following way

$$Z_{S^{d-1} \times S_\beta} = \sum_\Delta e^{-\frac{\beta}{R}\Delta}, \tag{4.7}$$

where $R$ is the radius of the sphere $S^{d-1}$. In the large volume *i.e.* $R \to \infty$ limit, if we probe the theory at a distance scale much larger than $\beta$, then we can describe it approximately by a local field theory on $S^{d-1}$ (assuming there is no symmetry breaking in the large volume limit). The effective Lagrangian should be constructed using local functions of the metric of the sphere $S^{d-1}$,

$$\sqrt{g}\mathcal{L} = \alpha_{d-1}\beta^{-d+1}\sqrt{g} + a_{d-2}\beta^{-d+3}\text{Ricci}\sqrt{g} + \cdots. \tag{4.8}$$

Thus in the large $R$ limit (which is equivalent to the fixed $R$, high temperature, $\beta \to 0$ limit), we have

$$\log Z_{S^{d-1} \times S_\beta} \sim a_{d-1}\beta^{-d+1}R^{d-1} + \#\beta^{-d+3}R^{d-3} + \cdots. \tag{4.9}$$

If we do a non-identity $g \neq e$ insertion then in this limit, we can view it as a defect from the point of view of the local field theory on $S^{d-1}$. One would expect that $a_{d-1}$ would decrease due to the defect (on account of the increase in the ground state energy of the defect Hilbert space due to the defect) and thus $Z(\beta, g)$ would be exponentially suppressed compared to $Z(\beta, e) = Z(\beta)$ for $g \neq e$.

## Acknowledgements

W. C. is supported by the Global Science Graduate Course (GSGC) program of the University of Tokyo, the World Premier International Research Center Initiative (WPI) and acknowledges support from JSPS KAKENHI grant number JP19H05810, JP22J21553 and JP22KJ1072. T. M. is supported by the World Premier International Research Center Initiative (WPI) MEXT, Japan, and by JSPS KAKENHI grants JP18K13533, JP19H05810, JP20H01896, and JP20H00153. S. P. acknowledges funding provided by Tomislav and Vesna Kundic as well as the support from DOE grant DE-SC0009988. We thank D. Harlow and H. Ooguri for valuable comments and feedback on the draft. S. P. thanks D. Mazáč for teaching him amusing facts about group theory, G. J. Turiaci for an insightful journal club talk on [17] and J. Magan for explaining his proof of the conjecture.

## A Symmetry resolution of weakly coupled free field theory in Canonical ensemble

As the mass dimension of an operator $\Delta$ becomes large, the number of such operators with dimension $\Delta$ grows roughly like $\exp\left[\#\Delta^{(d-1)/d}\right]$. The precise asymptotic growth may be derived utilizing Hilbert series and Meinardus' theorem, as shown in Ref. [16]. The Hilbert series

techniques coupled with Meinardus' theorem provide us with a partition function $Z_{\text{Free}}(\beta)$ (or a projection onto some subset of it) for the free theory at some inverse temperature $\beta$,

$$\log Z_{\text{Free}}(\beta \to 0) \sim a_{\text{free}}\beta^{-d+1} + O(\beta^{-d+2}). \tag{A.1}$$

Before we turn on interactions, there are no operators at some non-integer scaling (mass) dimension. When we turn them on, the operators obtain anomalous scaling (mass) dimensions and spread out away from the integer points. Thus one can not obviously write a Hilbert series for an interacting CFT using the conventional Plethystic exponential. However, we may still ask whether we can make some progress in understanding the high temperature behaviour of partition function of weakly interacting CFT. Intuitively the answer is yes if there is really a weakly coupled description and the anomalous dimensions of the operators are not too large: as long as the spreading is well controlled and bounded in an order one interval, we do not expect the high temperature behaviour to change.

In particular, under the assumption that the high temperature twisted partition function of weakly coupled QFT can be approximated by a Plethystic exponential form

$$Z(\beta, g) = \exp\left[\sum_{n=1}^{\infty} \frac{1}{n} \left(\chi_\gamma(g^n) + \chi_\gamma^*(g^n)\right) I(n\beta)\right], \tag{A.2}$$

where $\chi_\gamma$ is the character of the irrep $\gamma$; $I(n\beta)$ is a single particle partition function, using the arguments presented in section 2, we can immediately show the following.

**Canonical, Weakly coupled QFT:** For a sequence **T** of weakly coupled QFTs on $\mathcal{M} = S^{d-1}$, for small $\beta$, we can make $\lambda$ sufficiently small (this is a simultaneous limit, see the figure 1) such that we have

$$Z_\alpha(\lambda, \beta) \underset{\lambda,\beta \to 0}{\sim} \frac{\dim(\alpha)^2}{|G|} Z(\lambda, \beta). \tag{A.3}$$

This is the canonical version of the theorem 1.4.

In the rest of the appendix, we provide evidence that in the high temperature limit, $\log Z(\beta)$ behaves like the free theory partition function (and the same for $\log Z_\alpha$). To proceed, we recall eq. (3.1)

$$\int_0^\infty d\Delta' \, \Theta\left[\Delta' \in (n-1/2, n+1/2)\right] \rho(\lambda, \Delta') = d_n \tag{A.4}$$

At this point, our aim is to relate the L.H.S to the partition function of weakly interacting QFT, and to relate the R.H.S to the $Z_{\text{free}}(\beta)$ in the $\beta \to 0$ limit for $n = \lfloor \Delta_{\text{Saddle}}(\beta) \rfloor$, where $\Delta_{\text{Saddle}}(\beta)$ is obtained by solving $\langle T_{00} \rangle_\beta = \langle \Delta_{\text{Saddle}} | T_{00} | \Delta_{\text{Saddle}} \rangle$ for free theory i.e solving an equation which relates the microcanonical ensemble average of the stress energy tensor $T_{00}$ with the canonical ensemble average of $T_{00}$ for the free theory. In what follows, we will work with this choice of $n$ unless otherwise mentioned. Using the results of [16], we can show that in the $\beta \to 0$ limit, we have on the R.H.S.

$$d_n = Z_{\text{free}}(\beta)h(\beta)e^{\beta n} \tag{A.5}$$

where $h(\beta)$ is a polynomial in $\beta$. The explicit form of $h(\beta)$ is not very important, however it can be found in [16].

Manipulating the L.H.S. is more involed and our method is based on the same techniques as employed in [15]. We start with a majoriser $\phi_+$ and minoriser $\phi_-$ of the indicator function of the interval $[\Delta - \delta, \Delta + \delta]$ and $(\Delta - \delta, \Delta + \delta)$ respectively, here $\Delta = n, \delta = 1/2$:

$$\phi_-(\Delta') \leq \Theta(\Delta') \leq \phi_+(\Delta'). \tag{A.6}$$

These functions are also chosen to be bandlimited functions i.e. the support of their Fourier transform is bounded $(-\Lambda, \Lambda)$;

$$\phi_\pm(\Delta') = \int_{-\Lambda}^{\Lambda} dt \ \widehat{\phi}_\pm(t) e^{-i\Delta' t}. \tag{A.7}$$

Such functions occur in the modular bootstrap literature, see in particular Ref. [15]; we supply the relevant functions in the later part if this appendix and refer the reader curious about the details of their modular bootstrap applications to the reference. The zero modes of these functions are given by (see eq. (A.22))

$$2\pi\widehat{\phi}_\pm(0) = 2\delta \pm \frac{2\pi}{\Lambda}, \tag{A.8}$$

which will appear in our analysis below. With this machinery, as a first step, we show that

$$
\begin{aligned}
e^{\beta(n-1/2)} & \int_{-\Lambda}^{\Lambda} dt \ Z(\lambda, \beta + it)\widehat{\phi}_-(t) \\
& \leq \int_0^\infty d\Delta' \ \Theta\bigg(\Delta' \in (n - 1/2, n + 1/2)\bigg)\rho(\lambda, \Delta') \\
& \leq e^{\beta(n+1/2)} \int_{-\Lambda}^{\Lambda} dt \ Z(\lambda, \beta + it)\widehat{\phi}_+(t)
\end{aligned}
\tag{A.9}
$$

To derive the above, we start from eq. (A.6) and we write (recall $\Delta = n$ and $\delta = 1/2$)

$$e^{\beta(\Delta-\delta)} \int_0^\infty d\Delta' \ \rho(\lambda, \Delta')e^{-\beta\Delta'}\phi_-(\Delta') \leq \int_{\Delta-\delta}^{\Delta+\delta} d\Delta' \ \rho(\lambda, \Delta') \leq e^{\beta(\Delta+\delta)} \int_0^\infty d\Delta' \ \rho(\lambda, \Delta')e^{-\beta\Delta'}\phi_+(\Delta'). \tag{A.10}$$

In Fourier domain, the integrals appearing as the lower and the upper bounds can be recast as

$$e^{\beta(\Delta\pm\delta)} \int_{-\Lambda}^{\Lambda} dt \ Z(\lambda, \beta + it)\widehat{\phi}_\pm(t), \tag{A.11}$$

where $Z(\lambda, \beta + it) = \int d\Delta' \ \rho(\lambda, \Delta')e^{-(\beta+it)\Delta'}$. Thus we need to estimate

$$\int_{-\Lambda}^{\Lambda} dt \ Z(\lambda, \beta + it)\widehat{\phi}_\pm(t). \tag{A.12}$$

To proceed, we make an assumption on a quantity known as the spectral form factor. The spectral form factor (SFF) is given by $|Z(\lambda, \beta + it)|^2$, and in more conventional usage it

constitutes an important diagnostic of chaos. The SFF always has a peak at $t = 0$ where phases coherently add up. At non-zero times $t$, the phases start to cancel each other, we expect a sharp decay in the SFF, especially at high temperature. In two dimensions, that this happens can be shown using modular invariance [15]. In higher dimensions, this can been shown using the results of [16] for the free theory. For a weakly interacting theory, we put it in as an assumption. Mathematically, we require that $\beta, \lambda \to 0$ limit

$$\int_{-\Lambda}^{\Lambda} \mathrm{d}t \; Z(\lambda, \beta + it)\widehat{\phi}(t) \sim Z(\lambda, \beta)f(\lambda, \beta)\widehat{\phi}(0) \,, \tag{A.13}$$

for a polynomially behaved function $\widehat{\phi}(t)$ and sufficiently large $\Lambda$, here $f(\lambda, \beta) \geqslant 0$ and captures the fluctuation around the saddle at $t = 0$ and has a power law behavior in $\beta$. In practice, we want $\Lambda$ to be at least an order one number, greater than $2\pi/\delta$. This assumption is expected to be true for all theories irrespective of weak coupling. In 2D CFT, this in fact is proven [15]. Using this we obtain

$$Z(\lambda, \beta)f(\lambda, \beta)\widehat{\phi}_-(0) \leq e^{-\beta n} \int_{n-1/2}^{n+1/2} \mathrm{d}\Delta' \; \rho(\Delta') \leq Z(\lambda, \beta)f(\lambda, \beta)\widehat{\phi}_+(0) \,. \tag{A.14}$$

Combining eqs. (A.4), (A.5) and (A.14), we find that

$$Z(\lambda, \beta)f(\lambda, \beta)\widehat{\phi}_-(0) \leq Z_{\mathrm{free}}(\beta)h(\beta) \leq Z(\lambda, \beta)f(\lambda, \beta)\widehat{\phi}_+(0) \tag{A.15}$$

or equivalently,

$$\left(\frac{h(\beta)}{f(\lambda, \beta)\widehat{\phi}_+(0)}\right) Z_{\mathrm{free}}(\beta) \leq Z(\lambda, \beta) \leq \left(\frac{h(\beta)}{f(\lambda, \beta)\widehat{\phi}_-(0)}\right) Z_{\mathrm{free}}(\beta) \tag{A.16}$$

The above implies that in the $\beta \to 0$ and $\lambda \to 0$ limit,

$$\log Z(\lambda, \beta) \sim \log Z_{\mathrm{free}}(\beta) = a_{\mathrm{free}}\beta^{-d+1} \tag{A.17}$$

Since the leading term controls the fluctuation $h(\beta)$ in the case of free theory, and the fact that the leading term does not change for sufficiently weak coupling as evidenced by (A.17), we must have $h(\beta) = f(\lambda, \beta)$ in this limit. Hence it follows that $Z(\lambda, \beta)$ behaves like $Z_{\mathrm{free}}(\beta)$ up to an order one multiplicative number. We remark that we lose control over the order one multiplicative number this way. For eq. A.3 to follow for weakly coupled theory we have to appeal to the physical argument that at high temperature, we have a behavior similar to free theory, hence one is allowed to express $Z(\lambda, \beta, g)$ in the Plethystic form, given by eq. (A.2).

Let us mention how the argument leading to eq. A.3 can be modified to prove a result relevant for an arbitrary compact spatial manifold. The key point is to formulate all the statements above in terms of energy eigenstates on the manifold instead of using the state operator correspondence and writing things in terms of operator dimension and/or anomalous dimension. In this language, instead of anomalous dimension, we consider the shift in energies as a result of perturbation. The weakly interacting theory on a general manifold then behaves like the free theory on that manifold, and the free theory on an arbitrary manifold does satisfy eq. A.3.

### A.1 Beurling-Selberg functions

Here we reproduce the salient equations from [15] and explicitly state how to obtain useful functions by appropriate scaling of the functions. We start with $\phi_\pm(x)$ which majorizes and minorizes $\Theta(x)$ i.e

$$\phi_-(x) \le \Theta_{x \in (-\delta_*, \delta_*)} \le \Theta_{x \in [-\delta_*, \delta_*]} \le \phi_+(x) \tag{A.18}$$

where $\Theta_{x \in (-\delta_*, \delta_*)}$ is the characteristic function of the interval $(-\delta_*, \delta_*)$ and $\Theta_{x \in [-\delta_*, \delta_*]}$ is the characteristic function of the interval $[-\delta_*, \delta_*]$.

Explicitly we have

$$\begin{aligned}
\phi_+(x) &= \frac{1}{2} B_+(\delta_* + x) + \frac{1}{2} B_+(\delta_* - x)\,, \\
\phi_-(x) &= \frac{1}{2} B_-(\delta_* + x) + \frac{1}{2} B_-(\delta_* - x)\,,
\end{aligned} \tag{A.19}$$

where $B_\pm(x)$ are given by

$$\begin{aligned}
B_+(x) &= \frac{2\sin^2(\pi x)}{\pi^2} \left[ \sum_{k=0}^{\infty} \frac{1}{(x-k)^2} + \frac{1}{x} \right] - 1\,, \\
B_-(x) &= \frac{2\sin^2(\pi x)}{\pi^2} \left[ \sum_{k=1}^{\infty} \frac{1}{(x-k)^2} + \frac{1}{x} \right] - 1\,.
\end{aligned} \tag{A.20}$$

These functions have support $[-2\pi, 2\pi]$ in the Fourier domain and following zero mode

$$2\pi \widehat{\phi}_\pm(0) = 2\delta_* \pm 1\,. \tag{A.21}$$

Now we need to rescale these functions appropriately so that the support becomes $[-\Lambda, \Lambda]$ and the interval they majorize and minorize becomes $[-\delta, \delta]$ and $(-\delta, \delta)$ respectively. Then we need to set $x = \Delta - \Delta'$ to satisfy the eq. (A.6). The zero modes of these scaled functions are given by

$$2\pi \widehat{\phi}_\pm(0) = 2\delta \pm \frac{2\pi}{\Lambda}\,, \tag{A.22}$$

which appears in our analysis in the previous appendix.

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
