# Peer review of "Universal fine grained asymptotics of weakly coupled Quantum Field Theory"

_SciPost Physics_

## Round 2 · Referee Report · Anonymous (Referee 1) · 2023-5-10

Report

This paper studies the density of states and partition function for weakly coupled QFTs on a compact spatial manifold with finite group symmetry $G$. The main result is that the former at high energy and the latter at high temperature can both be projected to a particular representation $\alpha$ with a simple factor of $\text{dim}(\alpha)^2 / |G|$. This proves a conjecture which was originally argued from a gravitational perspective. My main concern is that a couple parts of the paper might lead readers to either underestimate or overestimate how general the results are.

Theorem 1.5 states that weak coupling needs to be suitably defined but (weaker) assumptions about what weak coupling means are already required for theorem 1.4. Moreover, fixed coupling appears to violate them. Looking at p10, one needs (29) to hold for $\beta = 0$ in order to claim that theorem 1.4 has been proven. The point underneath about taking $\lambda, \beta$ to 0 at the same time is therefore essential. Only special theories would allow the operators to stay away from each other's bins without this. As a result, theorem 1.4 seems to be a statement about a sequence of QFTs which become arbitrarily weak. I.e. for every $E$, there exists a point in this sequence beyond which equation (7) holds. Since this is an important distinction, I think it should be stated already in theorem 1.4 so that ``weak coupling is suitably defined'' in theorem 1.5 refers only to the additional assumption needed there. Namely the one about the decay of the spectral form factor.

A second point is that "anomalous dimension" appears to be only used as a mnemonic. In some examples, like the Wilson-Fisher and 3-state Potts models discussed in the paper, it is indeed most useful to study energies on $S^{d - 1}$ in order to learn about scaling dimensions of the CFT on flat space. However, I do not see any place where the state-operator map was really used in the proof. The strategy mentioned at the end of section 3.2, which involves integrating over energy eigenvalues of a potentially non-conformal QFT, seems to be what was done all along. If that is correct, there could be an earlier comment about how the derivation is general despite the $\Delta$ notation and analogy to CFT being convenient.

I hope clarifications along these lines are not too time consuming. While they are being made, the typos listed below should also be fixed.

Requested changes

  1. Fix spelling of "rationale" on p4, "greater" on p12, "lose" on p13 and "degeneracy" in (49).
  2. "Theorem 1.5" on p6 and "Bosonic" on p9 should not be capitalized.
  3. There should be no "s" in "finite groups" on p4 and "holds true" on p9.
  4. Equation (14) should say "the character orthogonality theorem" before and have a period after.
  5. I would say "that of the free theory" on p9 and "we want $\Lambda$ to be" on p12.
  6. A few places that say "$\beta \to 0$ limit" should say "the $\beta \to 0$ limit".
  7. I would make "1+1D", "1+1 D", "1+1-D" and "(1+1)-D" consistent.
  8. There are four equations without a number on pages 11, 12, 12 and 16.

  • validity: high
  • significance: high
  • originality: good
  • clarity: good
  • formatting: reasonable
  • grammar: reasonable

Author:  Weiguang Cao  on 2023-05-26  [id 3685]

(in reply to Report 1 on 2023-05-10)
Category:
correction

We thank the referee for the report. We agree with the suggested clarifications and requested changes, and they will be implemented in the resubmitted manuscript.

---

## Round 2 · Referee Report · Anonymous (Referee 2) · 2023-5-12

Report

In the present paper the authors consider free and weakly coupled QFTs with a finite group global symmetry G. They focus on high-energy eigenstates and ask what fraction of these states falls into a particular irrep $\alpha$ of G. The main goal is to prove the conjecture by Harlow-Oguri that this fraction is universal and determined by the dimension of the representation and the group order. This conjectured fraction is $dim(\alpha)^2 \over |G|$, eq. (1).

The part that is of some interest is the discussion of free CFTs (in any dimension d) in section 2. The main assumption is the existence of Fock space with one- and multi-particle states. As far as I understand, these results are known in the literature (including papers by some of the authors). Here, they give a bit more detail and estimate the corrections to show that they are suppressed.

The main part of the paper (section 3) is about weakly coupled QFTs, a small perturbation on the free CFT. This part of the paper I did not find very convincing. It seems that the argument is essentially to scale the coupling to zero, as we scale the energy to be high, in whatever way that is needed to still rely on free CFT results. There are no quantitatvie estimates or examples on how exactly we need to scale the coupling. (They consider some examples on what happens with a particular operator at weak coupling in section 3.3, but not the density of states). In the end, I did not find any results that go beyond the free case, except for some general words that the coupling is small and things work out roughly as in free CFT.

In the micro-canonical case, section 3.1, they take the coupling small enough such that all anomalous dimensions are very small and any given window of energies contains essentially the same operators as in the free CFT. Again, the argument is to say that it's the same as in free CFT.

In the canonical case, section 3.2, they make some estimates on the partition function, but finally in the paragraph after eq. (42) seem to admit that they do not have control over the term responsible for $dim(\alpha)^2 \over |G|$.

Naively, it seems obvious that perturbative corrections to the partition function are of the form $Z(\beta) = Z_{free}(\beta) (1 + O(\lambda))$, so I'm not sure why one has to go through the estimates (32) - (42) in the first place.

In light of the above, I find it difficult to recommend this paper for publication. At least not without some substantial revision/expansion of the main arguments and/or examples. Or perhaps I missed something important and the authors can explain their arguments in more detail.
  • validity: -
  • significance: -
  • originality: -
  • clarity: -
  • formatting: -
  • grammar: -

Author:  Weiguang Cao  on 2023-05-26  [id 3686]

(in reply to Report 2 on 2023-05-12)
Category:
answer to question
correction

We thank the referee for the report.

In response:

We wish to clarify that the free theory result is not known in the literature. This is new in this paper, and non-trivial. The free theory results are the main technical result of this paper. We then use these results to make some further statements about weakly coupled QFT. We accept that the free theory result was not emphasized enough in the submitted version. We would make this clearer in a resubmitted version.

In response to
-Naively, it seems obvious that perturbative corrections to the partition function are of the form
Z(β) = ZFree(β) [1 + O(λ)] , (7)
, so I'm not sure why one has to go through the estimates (32) - (42) in the first place.

Naively one would expect that Z(β) = ZFree(β) [1 + O(λ)] , where λ is the coupling parameter (appropriately made dimensionless by a relevant energy scale, for example μ). While the above equation is true for finite β, the subtlety lies in the fact that the order of λ estimate is not necessarily uniform in β. In other words, the correction term O(λ) can potentially depend on β and might get enhanced as we take the β → 0 limit. This necessitates that we take a simultaneous limit β → 0 and λ → 0 i.e we are scanning over a sequence of weakly coupled theories such that they become arbitrarily weak as β → 0.

See also response to report 3 regarding the meaning of sequence of weakly coupled theories.

In response to
-There are no quantitative estimates or examples on how exactly we need to scale the coupling.
And
- In the end, I did not find any results that go beyond the free case, except for some general words that the coupling is small and things work out roughly as in free CFT

As above, the free theory is the main technical result, and then we are making further statements about weakly coupled QFT. In section 3, we make precise statements about how the coupling needs to approach zero, which depends on the form of the anomalous dimensions of the operators in the particular theory. This is quantitive in the sense that for a given theory, with given anomalous dimensions of some operators, we have a result on how the coupling should be taken to zero.

---

## Round 2 · Referee Report · Anonymous (Referee 3) · 2023-5-17

Report

This paper aims to prove certain theorems about the asymptotic density of states of QFTs in general dimension in the presence of global symmetry. Unfortunately, I found the paper to fall short of this goal, due to a significant lack of precision. There are many loose ends to the arguments - unclarity both about what is being assumed, and what the theorems even are supposed to be - such that altogether, nothing is "proven" in any meaningful sense of the word. There are significant issues of clarity: in the statement of the conjectures and theorems themselves; in the approach to proving them; and in the assumptions that such attempts employ.

On top of that, there is a general lack of attention to details, both in terms of the physical arguments and the writing of the paper itself. Typos and half-hearted arguments abound. Overall it was a quite frustrating read, especially because I feel the theorems are likely to ultimately be correct.

One of the main issues revolves around the lack of any clear notion of "weak coupling". There is something absurd about stating a theorem as in Theorem 1.5 (i): in particular, including the phrase "where the weak coupling is suitably defined." But this definition is at the crux of the topic! And the main text itself makes no such suitable definition. (Indeed there are contradictory statements in the paper about what "weak coupling" is supposed to mean, as we discuss below.) As such the theorem stated in this way has very little content: a theorem should be sharply stated.

Let us dive into the topic of the meaning of weak coupling. The main ambiguity is whether the authors mean infinitesimally weak coupling -- i.e. perturbatively close to a free theory -- or small but finite coupling. These are very different. (For example, the spectrum of the former is non-chaotic, whereas the spectrum of the latter would be expected to be chaotic.)

From Theorems 1.3 and 1.4, the authors seem to be restricting themselves to the perturbative regime: otherwise, there is no notion of a "fundamental field." Around eq. 29 a first attempt at a definition of weak coupling is given, in which a certain set of operators is taken to have "tiny anomalous dimension." I hope we can agree that this is not a well-defined concept. Eq. 29 and the words below try to clarify, but it is not at all clear what is going on with respect to the limit being taken. (Is eq 29 supposed to hold for nonzero-but-small $\epsilon$? etc.) Accordingly the claim of proof below eq. 31 doesn't hold up, because "weakly coupled QFT" is not yet well-defined.

We then keep reading and arrive at the longer discussion a few pages later, below Figure 1, about what the limit of $\beta$ and $\lambda$ is. Putting aside that $\lambda$ is in general dimensionful - so that both $\lambda$ and $\beta$ should be rendered dimensionless using the spatial scale $R$ before discussing taking a limit - we read that "for finite but small $\beta$, $\lambda$ is nonzero and thus we are actually probing weakly interacting QFT." This is no longer perturbative! With respect to the attempted proof, the assumptions about the stability of the number of operators in an energy interval of fixed size are not justified for finite coupling. So if the authors actually mean finite $\lambda$, the assumptions about the spectrum being used are seriously called into question. At the very least, the authors should have made clearer what they even mean, at a quantitative level beyond what is done in the paper.

Then at the bottom of p.13/top of p.14, the generalization to Theorem 1.5 (ii) is claimed to be done in a few sentences. Again, this does not suffice.

Perhaps one can summarize the above as saying that the authors do not make at all clear 1) what it means to be sufficiently close to a free theory, nor 2) whether the assumptions that they wish to use, key as they are to the whole point of the paper, are well-justified physically. The "proofs" just shift the original questions to these questions; but they are not properly analyzed.

Further comments:

  • An assumption about the spectral form factor enters. This further dilutes the claimed proofs.

  • I was not sure why the theorems were stated for spheres and other spatial manifolds separately.

  • The purpose of Section 3.3 is unclear (it does not manifestly demonstrate the theorems).

  • The leap to Conjecture 1.6 is obviously up to the authors but given its generality it has the feeling of coming essentially out of nowhere with respect to the rest of the paper (this is of course a subjective comment).

  • As mentioned earlier, there is an overall sense of sloppiness, including many typos, inconsistent nomenclatures, and so on. The referencing was also rather incomplete.

There is more that could be said here, but given all of the factors, and the SciPost standards for publication, I feel it is necessary to reject the paper. (A major revision might shore up the proof for perturbative QFTs, but I do not feel that that result would meet the bar for publication.)

  • validity: low
  • significance: low
  • originality: -
  • clarity: -
  • formatting: below threshold
  • grammar: good

Author:  Weiguang Cao  on 2023-05-26  [id 3687]

(in reply to Report 3 on 2023-05-17)
Category:
answer to question
objection
correction
validation or rederivation

We thank the referee for the report.

In response to: - There are many loose ends to the arguments - unclarity both about what is being assumed, and what the theorems even are supposed to be - such that altogether, nothing is "proven" in any meaningful sense of the word. 

We disagree with the above statement, and wish to clarify that Section 2 (free theory) is the main technical result of this paper. Here we prove theorem 1.3, concretely. This result was was not previously known in the literature. We address the question about assumptions and wording of theorems below.

In response to -One of the main issues revolves around the lack of any clear notion of "weak coupling". There is something absurd about stating a theorem as in Theorem 1.5 (i): in particular, including the phrase "where the weak coupling is suitably defined." But this definition is at the crux of the topic! And the main text itself makes no such suitable definition. (Indeed there are contradictory statements in the paper about what "weak coupling" is supposed to mean, as we discuss below.) As such the theorem stated in this way has very little content: a theorem should be sharply stated.

The technical meaning of weak coupling is as follows. We consider a sequence of such theories labeled by Q_λ such that λ → 0. In this notation Q_0 is the free theory. The energy spectrum of the theory Q_λ is given by E(λ) with density of states ρ(λ, E). We further require that given energy E, we can make λ sufficiently small such that ρ(λ, E′) has perturbative description in λ for E′ < E and this continues to hold as we let E → ∞ with possibly tuning λ → 0. We will call such a sequence T. The sequence T can as well be a sequence of free theories. From now on by weakly coupled theory, we will mean the elements of the sequence T. We will modify the theorems to include this definition.

In response to -Let us dive into the topic of the meaning of weak coupling. The main ambiguity is whether the authors mean infinitesimally weak coupling -- i.e. perturbatively close to a free theory -- or small but finite coupling. These are very different. (For example, the spectrum of the former is non-chaotic, whereas the spectrum of the latter would be expected to be chaotic.)

We hope this is clarified by the above. We take lambda to zero limit. We agree that all phrases such as 'small but finite' that contribute to this ambiguity should be removed.

In response to -From Theorems 1.3 and 1.4, the authors seem to be restricting themselves to the perturbative regime: otherwise, there is no notion of a "fundamental field." Around eq. 29 a first attempt at a definition of weak coupling is given, in which a certain set of operators is taken to have "tiny anomalous dimension." I hope we can agree that this is not a well-defined concept. Eq. 29 and the words below try to clarify, but it is not at all clear what is going on with respect to the limit being taken. (Is eq 29 supposed to hold for nonzero-but-small epsilon? etc.) Accordingly the claim of proof below eq. 31 doesn't hold up, because "weakly coupled QFT" is not yet well-defined.

Thm 1.3 is free theory. Thm 1.4 is strictly perturbative, yes. For Eq 29 the use of \epsilon is superfluous. It can be restated as if the shift is small enough, which can be arranged by making $\lambda$ sufficiently small, we must have

$$ \int_{0}^{\infty}\mathrm{d}\Delta'\ \Theta\left[\Delta'\in (k-1/2,k+1/2)\right]\ \rho(\lambda,\Delta')=d_k\,,\quad k\leqslant N $$
where $\Theta\left[\Delta'\in (k-1/2,k+1/2)\right]$ is the characteristic function for the open interval $(k-1/2,k+1/2)$. As $N\to\infty$ if needed, we might have to take the coupling $\lambda\to 0$ limit to make sure that Eq.29 holds true.

In response to -We then keep reading and arrive at the longer discussion a few pages later, below Figure 1, about what the limit of β and λ is. Putting aside that λ is in general dimensionful - so that both λ and β should be rendered dimensionless using the spatial scale R before discussing taking a limit - we read that "for finite but small β, λ is nonzero and thus we are actually probing weakly interacting QFT." This is no longer perturbative! With respect to the attempted proof, the assumptions about the stability of the number of operators in an energy interval of fixed size are not justified for finite coupling. So if the authors actually mean finite λ, the assumptions about the spectrum being used are seriously called into question. At the very least, the authors should have made clearer what they even mean, at a quantitative level beyond what is done in the paper.

Indeed, lambda is taken dimensionless. All mention of “finite lambda” or “finite beta” should indeed be removed to make things more precise, and in accordance with our response above.

In response to -Then at the bottom of p.13/top of p.14, the generalization to Theorem 1.5 (ii) is claimed to be done in a few sentences. Again, this does not suffice. It was our purpose only to sketch how the generalziation to Thm 1.5 ii should be performed. We agree this could be de-emphasized further, for example by removing (ii) from Thm 1.4 1.5, and relegating this to a discussion of how one would apply these ideas to a more general manifold. For Thm 1.3, the argument is however concrete

In response to further comments - I was not sure why the theorems were stated for spheres and other spatial manifolds separately. Addressed above. - The purpose of Section 3.3 is unclear (it does not manifestly demonstrate the theorems). The purpose of this subsection is to provide examples which exhibit that the anomalous dimension of heavy operators can be bounded by an order one number provided we make the coupling sufficiently small. - The leap to Conjecture 1.6 is obviously up to the authors but given its generality it has the feeling of coming essentially out of nowhere with respect to the rest of the paper (this is of course a subjective comment). The reason for making the conjucture (Eq 14 ) follows from the ideas in eqs 10-13.

---

## Editorial Decision

rejected_or_withdrawn